# Relationship between East Asian Cold Surges and Synoptic Patterns: A New Coupling Framework

**Anupam Kumar [1,2,3,*], Edmond Y.M. Lo [2,4] and Adam D. Switzer [2,5,6]**

[1]   Interdisciplinary Graduate School, Nanyang Technological University, Singapore 639798, Singapore

[2]   Institute of Catastrophe Risk Management, Nanyang Technological University, Singapore 639798, Singapore;
      cymlo@ntu.edu.sg (E.Y.M.L.); aswitzer@ntu.edu.sg (A.D.S.)

[3]   Solar Energy Research Institute of Singapore, National University of Singapore, Singapore 117574, Singapore

[4]   School of Civil and Environmental Engineering, Nanyang Technological University,
      Singapore 639798, Singapore

[5]   Asian School of Environment, Nanyang Technological University, Singapore 639798, Singapore

[6]   Earth Observatory of Singapore, Nanyang Technological University, Singapore 639798, Singapore

*   Correspondence: anupam002@e.ntu.edu.sg; Tel.: (+65) 91239811; Fax: (+65) 6791-0676

**Abstract:** Strong cold surge events (CSEs) are some of the most distinct winter weather events in East Asia, impacting natural ecosystems and over 100 million individuals. The impact of such extreme CSEs as driven by synoptic systems is direct and immediate. Changes in large-scale synoptic patterns as potentially affected by changes in the Arctic are further expected to influence CSE occurrences in East Asia. Defying a straightforward analysis, semi-permanent atmospheric systems such as the Siberian High (SH), influencing large-scale synoptic patterns, make the atmospheric circulation highly variable and assessment of CSE onset difficult. Rather varied region-specific metrics are currently adopted for predicting CSE occurrence locally but the fundamental understanding of the onset of CSEs continues to be a major challenge. Based on an analysis of monthly synoptic patterns for three unusual CSEs in East Asia and further extended for eight strong to extreme CSEs, we propose a new coupling framework for an improved understanding and interpretation of the atmosphere dynamics driving CSE onset. The coupling framework involves linkages between the Siberian High, Aleutian Low, and Jet Stream. We also present the first meteorological scale for categorizing the intensity of such unusual CSEs.

**Keywords:** cold surge; synoptic patterns; Siberian High; Aleutian Low; Jet Stream

## 1. Introduction

Some of the most damaging weather events over Asia are cold surges [1] that occur during the cold season (Northeast Monsoon/winter monsoon) from November to February. Strong cold surges have killed more than 129 people and have contributed to direct economic losses exceeding US$20 billion [2]. The onset of cold surge events (CSEs) in East Asia is statistically linked with the intensity of the Siberian High (SH) and its southeast propagation [3,4]. While they occur once or twice monthly during the cold season and may last from a few days to a week or even longer [5,6], some surges can be particularly damaging. Strong CSEs are associated with snowfall and freezing precipitation over East Asia [2,4,7,8] and are the root cause of the severe weather over the South China Sea (SCS) [9], exerting tremendous economic and societal impacts. For instance, an extreme CSE that occurred from 10 January to 5 February 2008 induced very damaging ice storms, snow, and frost in South China [10,11] and caused US$24 billion in economic losses [12,13]. While CSEs majorly affect the convective activities over Asia, recent research further shows that they also impact on the tropical rainfall distribution, reaching far to the Indonesian maritime continent in Southern Hemisphere [14–16].

It is reported that climate change is "perhaps the single greatest challenge confronting the Asia-Pacific region, and its more than 4 billion people [17]". A recent study has indicated that SH intensity is enhancing [18], favoring a southward intrusion of cold air towards East Asia [19], though global warming trends are also expected to affect the SH. The warming trend over the period 1901–2009 is particularly strong for the semiarid area of Asia, with an increase of 2.4 °C during the cold season between November and March. It has been observed that the warming trends over the continental Asia are the strongest and particularly from 1979 onwards, the warming is strongest for northern and eastern Asia in autumn and spring, respectively, and for China in winter [20].

With regard to CSEs, here we show how the intensity of SH will influence CSEs over East Asia is not the only key issue; rather there are additional large-scale synoptic systems that together with the SH govern the onset and trajectory of CSEs. Studies on assessing such linkages for CSE occurrence (the potential of SH amplification on influencing large-scale synoptic pattern in East Asia is one such linkage) mostly proposed a single causal pathway. Examples include frequency of cold surges with respect to the El Niño–Southern Oscillation [21], the influence of Arctic Oscillation on cold surge occurrence [22], and the role of Sea Surface Temperature anomalies in cold events [23]. However, such singular linkage pathways are unlikely due to the complexity of the atmospheric dynamics. Thus, reported works on CSE occurrences focus on such classical approaches describing a single event as being based on a localized criteria and with the event evolving over a period of around one week. Local temperature drops over 24 to 48 hours, and combined with minimum wind speed at some locations, are used at, for example, the Hong Kong Observatory (HKO) and by the Korean Meteorological Administration, Japan Meteorological Agency, and others for assessing CSE occurrences. However, there are often consecutive signals of CSE over the northern SCS where the surges appear to be series of sequential events with in-between surge duration of a few days (e.g., CSEs of December 2005, January–February 2008, and January 2011). The inability to explain these CSEs along with failures in detecting the possible underlying relationships are at times interpreted as no linkage or evidence against linkages.

Here we show that the physical process of the upper troposphere need to be considered as both the lower and upper troposphere are likely related when explaining CSE occurrence mechanisms. Three atmospheric systems that significantly contribute to the linkage are analyzed here:

(1)  Influence from a single semi-permanent system (anticyclonic high pressure system over Siberia)
(2)  Coupled semi-permanent systems (anticyclonic high pressure system over Siberia and cyclonic low pressure system over Aleutian Island)
(3)  Influence from the upper troposphere via simultaneous forcing by upper level easterly winds and formation of low pressure systems near the coast of Japan.

The potentially huge socioeconomic impacts caused by extreme CSEs motivate the important questions on the formation of these CSEs that the existing atmospheric models can provide. The new coupling framework developed here explains the simultaneous processes leading to the formation and development of CSEs in East Asia, thereby extending our understanding of such coupled systems. The framework is derived based on detailed analysis of three reported strong CSEs over East Asia during January of 2008, 2011, and 2016 using data from the European Centre for Medium Range Weather Forecast (ECMWF), ERA Interim datasets (https://www.ecmwf.int/en/research/climate-reanalysis). These three notable recent CSEs were chosen for our study due to their relatively large impact, recent occurrence and the availability of data. The analysis is extended for a further eight strong to extreme CSEs reported over Asia that resulted in severe socio-economic impacts as reported in the Emergency Events Data Base (EM-DAT) https://www.cred.be/projects/EM-DAT. To the best of our knowledge, we also propose the first meteorological scale for predicting the strength, intensity, and direction of the CSEs.

## 2. Possible Mechanism on the Outbreak of CSE

An inadequate representation and understanding of dynamical processes in atmospheric models substantially contribute to the large uncertainties in CSE onset, whether the approach used is targeted model simulations or statistical analyses. Many previous analyses for CSEs have been performed with a straightforward SH intensity approach. However, CSEs are often preceded by upstream upper-level disturbances originating in the western Eurasian continent [6,24]. Despite the evidence of this linkage, the possible influence from the upper troposphere to the mechanism of the cold air outbreaks has not been fully explored. While the usual approach of SH intensity has widespread use in predicting usual CSE occurrence (e.g. at the HKO), we propose a system-level framework that links the large-scale circulation patterns in the atmosphere in CSE evolution. In this framework are linkages between the three atmospheric systems as discussed next.

### 2.1. Influence from a Single System

SH amplification is recognized as an essential factor for the generation and maintenance of CSEs [6] The SH dominates much of the Eurasian continent and is maintained by large-scale subsidence of air masses and strong radiative cooling [25]. To establish the relationship between cold air outbreaks in winter and its progression towards lower latitudes, we first define the domain of SH as between 40 °N–60 °N and 80 °E–120 °E for assessing its intensity. While it is noted that various slightly different domain definitions have been used [26–28], this domain definition has also been previously adopted for the SH [29,30]. Changes in SH intensity in the last three decades are noticeable as shown in Figure 1 for the monthly averaged values over the period 1979–2016. For January, the strongest SH intensity with a Mean Seal Level Pressure (MSLP) of 1040 hPa occurred during January 2016. For December, the strongest MSLP of 1037 hPa occurred during December 2005, while for February, the strongest MSLP of 1035 hPa occurred in February 1988. It is further noted that amplified period of SH is more evident for the later period of 1999–2016, whereas the intensification is significantly lower in earlier period from 1979 to 1998. These observations suggest a recent increase in the SH amplification over Siberia. Furthermore, this amplifying period is also coincident with the reported warmer decadal period of 1999–2008 [31]. The decade 2001–2010 was the warmest decade on record [32] (Supplementary Table S1) with the recent years 2015, 2016, and 2017 being the three warmest years [33] (Supplementary Table S2). It has been recently reported that that Arctic warming is accompanied by a rapid decline of sea ice cover while Eurasia is cooling, and with this the SH is intensifying gradually [18]. This SH intensification is likely associated with the Arctic amplification during these months as related to albedo changes from loss of snow and sea ice, and the presence of heat-trapping clouds and water vapor [34,35]. The intensification of SH has a noticeable trend line for January (Figure 1) with spatial dominance of MSLP exceeding 1030 hPa in areal extent for January (Supplementary Figure S1). This shows that January is the most critical month, followed by December and February for the outbreak of the cold air masses travelling from Arctic towards Siberia.

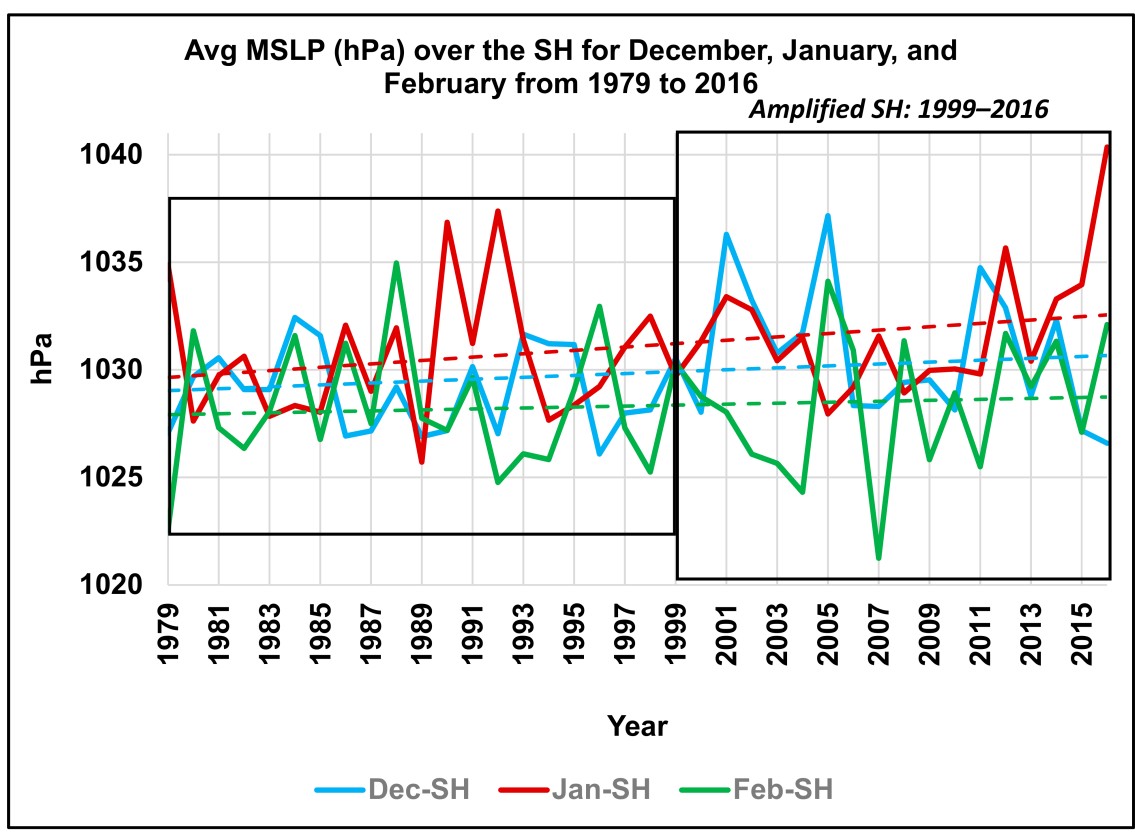

**Figure 1.** Average MSLP (hPa) over the Siberian High (SH) domain defined between 40 °N–60 °N and 80 °E–120 °E for December, January, and February from 1979–2016 as computed from the European Centre for Medium Range Weather Forecast (ECMWF), ERA-Interim. Dotted lines indicate fitted linear trends for December (blue color), January (red color), and February (green color) with only January showing a noticeable trend with a *p*-value of 0.06 from Mann–Kendall test.

Figure 2a–c illustrates three spatial configurations of the SH domain MSLP during the months of reported unusual CSEs. Figure 2a is for January 2008 that had a relatively widespread MSLP above the January climatological mean (over 1979–2016) of 1031 hPa. Figure 2b is for January 2011, exhibiting a more intensified MSLP with central pressure reaching up to 1045 hPa, followed by January 2016 (Figure 2c). Over the almost three decades (1979–2016), the three highest monthly MSLP values for January were 1040 hPa, 1037 hPa, and 1036 hPa for the years 2011, 2016, and 2008, respectively, with center of maximum MSLP shifted westward during 2008 and 2016, and shifted eastward during 2011 from its climatological average center (Figure 2d) located at about 50 °N, 100 °E. The westward shift of the central MSLP allow the cold winds to stream into China mainly from the west rather than via the usual northerly route. During a warmer Arctic period, a weaker polar vortex produces more northerly and less easterly winds at the mid-troposphere that has significant impact on surface winds in China [18].

However, regional variations of the cold winds streaming into China can be complex and sometimes counter-intuitive. For instance, the strongest cold air outbreaks over Siberian usually mean a strong CSE over East Asia. However, this outbreak (e.g. January 2016, January–February 2008) could also result in varying intensities at different locations due to changes in the atmospheric circulation. For example, increasing temperatures in the lower atmosphere elevates the mid-level pressure surfaces (geopotential height) resulting in wind pattern changes. The ability to predict the progression of cold air outbreak during CSEs then necessarily needs an understanding not only of thermal but also of the dynamical processes associated with it.

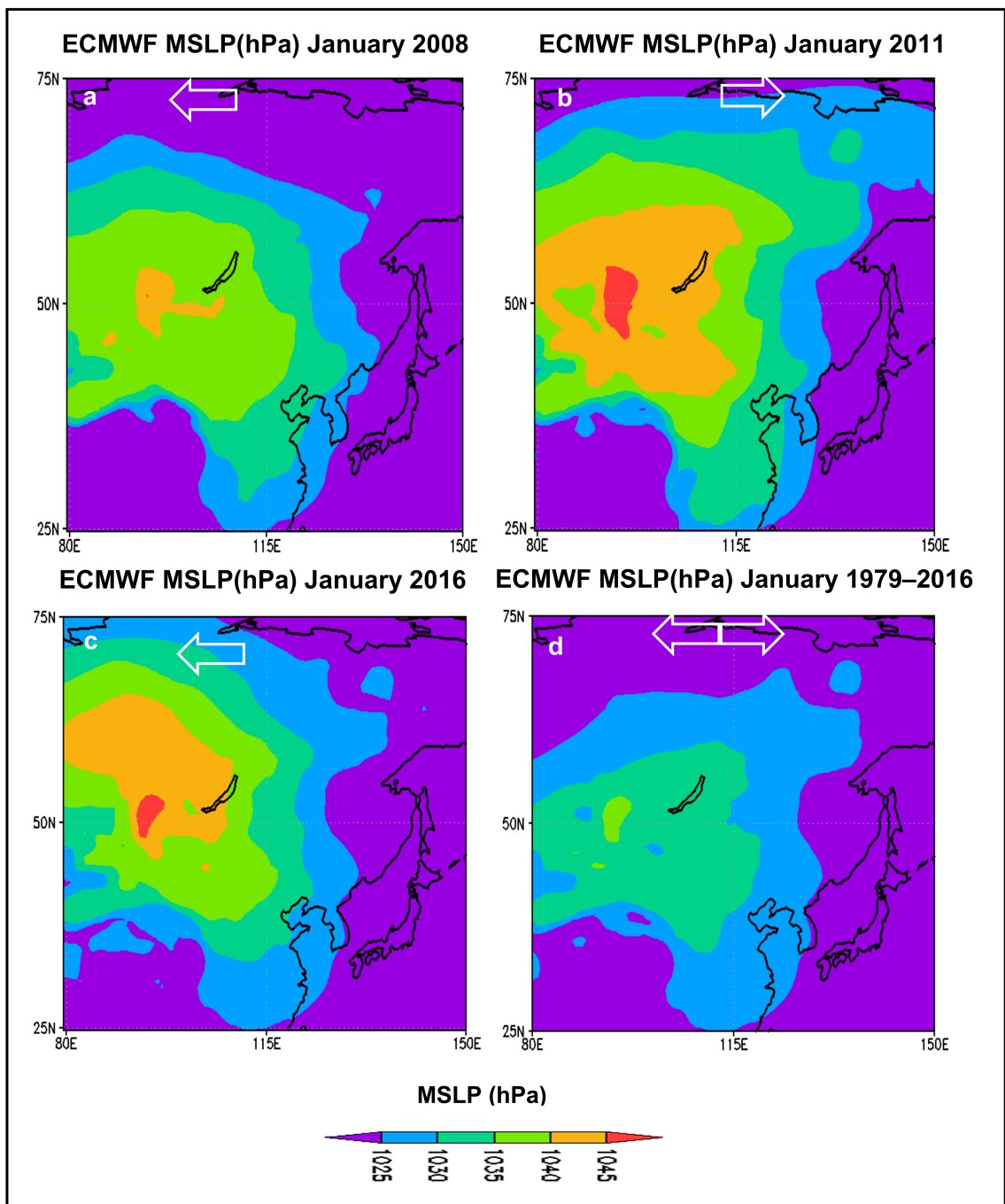

**Figure 2.** Monthly averaged MSLP over the SH for (**a**) January 2008; (**b**) January 2011; (**c**) January 2016; and (**d**) Annual mean MSLP for January over the SH for the period 1979–2016. White arrows indicate shifting of the SH system from the center (50 °N, 100 °E) of the domain. The coastal boundaries of Lake Baikal are represented by thick black lines. Data from the ECMWF, ERA-Interim.

## 2.2. Influence from a Coupled System

Changes in the circulation of the atmosphere and ocean are an integral part of climate variability and change [20]. The warmest temperature over land in the decade 2001–2010 was recorded in 2007, warmest temperature over ocean was in 2003, while the warmest temperature for both was recorded 2010 [32] (Supplementary Table S1). This is consistent with the current understanding of climate change science, which projects the ocean to warm at a slower rate compared to land and with much of the heat over land transported either into the ocean or to the atmosphere through evaporation.

Relating atmospheric circulation to the surface environment via atmosphere-ocean coupling is one of the fundamental objectives of synoptic meteorology [36]. One of the main climatological drivers situated in north Pacific is the Aleutian Low (AL), which plays a vital role in sea-air interaction. It is one of most intense (lowest) pressure system during the winter but practically disappears during the summer [37] and has the potential to influence the climate anomalies in remote regions [38,39].

Our hypothesis in overcoming the present limitation of correlating CSEs with only SH intensity is that other linkages, specifically the AL fundamentally alter progression of CSE itself. The surface weather over a region is the result of combined effects of local and large-scale forcing mechanisms. The pressure field is one of the factors that significantly influences atmospheric motion from the acceleration of wind from a high pressure region to low, i.e. the pressure difference (PD). For large-scale flows, apart from the PD there are other forces such as surface friction force (resulting in formation of Ekmann spiral), Coriolis force generally balancing the pressure difference, and centrifugal force. Here we analyze the outbreak of CSEs from the SH in relationship to the intensification of AL, considering them as coupled through the PD between the two systems. To quantify this PD during the winter season, we propose the domain of AL based on its climatology from 1979–2016 as between 25–75 °N and 160 E–130 °W.

A possible pathway for the linkage of the coupled SH–AL system during the outbreak of the unusual CSE in January 2016 is shown in Figures 3 and 4, which further illustrate the cause and effect of sequential events. The daily SH–AL PD value during January 2016 shows a sudden and steep increase from 24.9 hPa on 18 January at 12:00 UTC, lasting a continued 96 hours before rising rapidly to reach the month's maximum PD of 49.7 hPa on 22 January at 18:00 UTC (Figure 3). The CSE that streamed into China was recorded as the strongest during 21–25 January 2016. Figure 4 shows that progression of the intense cold air masses from higher towards lower latitudes is due to the intensification of AL, resulting in an enhanced PD and driving the winds from the Arctic Circle through Eastern Russia and towards the AL system. Such AL intensification (reduced MSLP) leads to an enhanced PD which leads to intense pressure gradient forming between SH and AL. This is evidenced through the MSLP plot, where the close isobars represent a large pressure change, thus driving the winds from the higher latitudes (Supplementary Figure S2). This PD drove a wide continuous band of cold winds with speed of 24–27 m/s (red arrow, Figure 4) moving consistently over the Arctic Circle through the North Pacific Ocean and entering the AL region at a speed of 18–21 m/s (green arrow, Figure 4). This synoptic circulation pattern resulted in a severe weather situation lasting a week over eastern Asia, resulting in the CSE of January 2016.

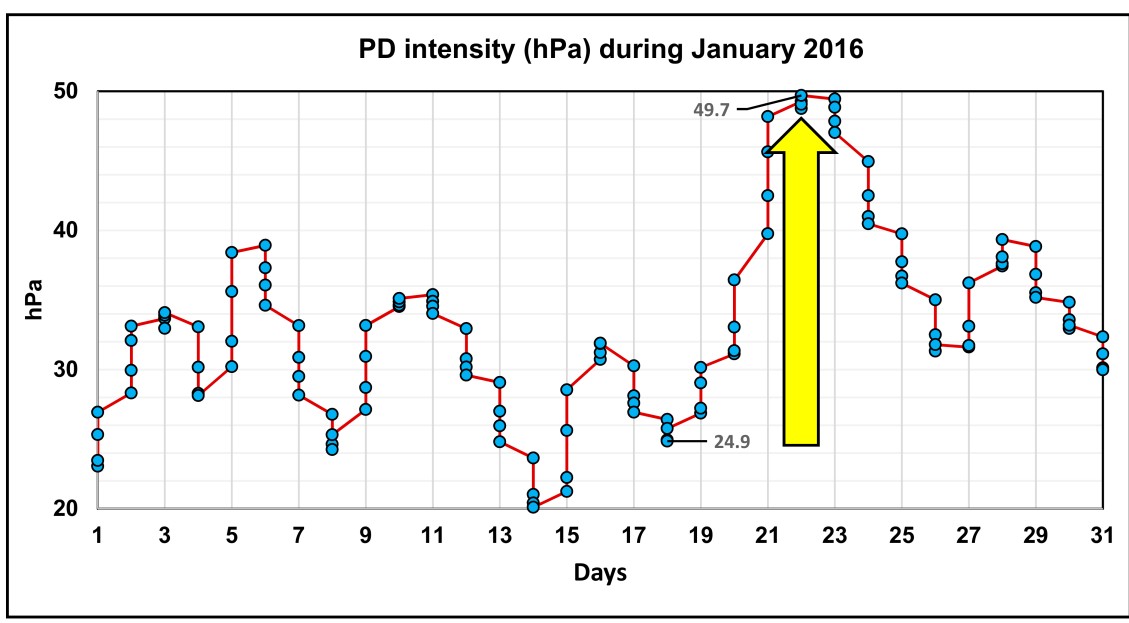

**Figure 3.** ERA-Interim pressure difference (PD; hPa) for January 2016 between the SH and Aleutian Low (AL) domains computed in 6-hourly intervals. Yellow arrow shows rising pressure difference (PD). Data from the ECMWF, ERA-Interim.

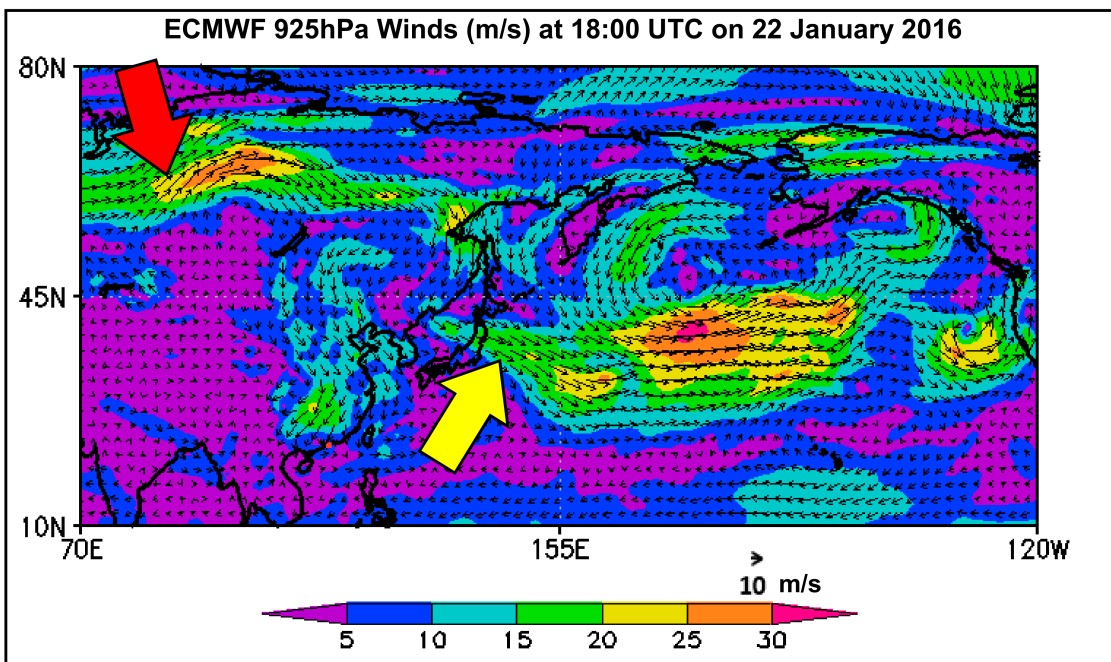

**Figure 4.** ECMWF 925 hPa winds (m/s) at 18:00 UTC 22 January 2016 over the SH and AL domain. Red and green arrows show progression of winds from Arctic Circle towards AL. Data from the ECMWF, ERA-Interim.

The CSE that occurred over January–February 2008 is the longest ever reported CSE that features a series of surges. In a recent study it was suggested that this CSE was tied to the roles of La Nina, North Polar Vortex and intra-seasonal oscillation for the stability of the surge event, but the results do not provide a definitive answer on the behavior of this unprecedented event [2]. For this event, the SH intensity remained almost consistent above 1030 hPa for nearly 35 days from 10 January to 14 February 2008 (Figure 5).

We categorized these 35 days into for different episodes based on the PD intensity between the coupled SHAL systems. The period from 9 to 15 January is defined as the first episode of CSE 2008 (Figure 5a). During this period the SH started intensifying continuously from 9 January and reached the month's maximum value of 1047 hPa on 12 January whilst the AL started intensifying from 12 January but weakened from 15 January. The intense SH but weakened AL led to a "stall" situation for the further progression of the cold air masses towards South China. We next define the period from 17 to 20 January as the second episode (Figure 5a). With the consistent intensified SH and AL, and with the latter dropping by 2 hPa, there was a gradual progression of cold air masses towards South China. The third episode is defined as the period from 24 to 29 January (Figure 5a). During this period, the AL continuously intensified with a steep MSLP drop of 9 hPa indicating strong progression of cold air towards South China. Lastly, the period from 3 to 18 February is defined as the fourth episode (Figure 5b). The SH further intensified during this period and reaches month's max MSLP of 1039 hPa on 11 February while the AL intensified with a steep drop in MSLP of 16 hPa reaching month's lowest MSLP of 1000 hPa on 18 February, indicating strong progression of cold air towards South China. This episode categorization is supported by Shi et al. (2010) [40] who similarly categorized the event in four episodes from 10 to 16 January (first episode), 18 to 22 January (second episode), 25 to 29 January (third episode), and 31 January–6 February fourth episode), though their categorization is based on four precipitation episodes that occurred as accompanied by sustained low temperatures. Notably our categorization of the fourth episode being from 3 to 18 February is longer than that by Shi et al. which was from 31 January to 6 February, this providing for a better explanation of the progression of the stalled surges in central China towards South China. Thus the HKO recorded its longest persisting cold spell since 1968, with a duration from 24 January to 16 February 2008.

The Jet Stream (JS) over East Asia is another important large scale circulation system of the atmosphere that significantly influences weather and climate over the Asian–Pacific region. For assessing the JS effects, we define the JS domain as between Lat 25 °N–5 °N and Lon 70 °E–180 °E, i.e., stretching from East Asia to the northern Pacific with a peripheral wind moving eastward. We further define the inner core of the JS referred herein as a "jet streak" as extending from around 130 °E up to 180 °E in longitude and confined between 25 °N to 45 °N in latitude. Our analysis suggests a monthly climatological (1979–2016) variation of the JS over East Asia during the Northeast Monsoon that is most dominant for January followed by February and December, as based on the longitudinal extent of the embedded jet streak (Supplementary Figure S3 for JS intensity at 250 hPa).

Before the outbreak of CSE of January2016, the intensity of the JS started strengthening for 8 consecutive days, increasing from 52.0 m/s on 19 January at 00:00 UTC to reach the month's maximum value of 67.7 m/s on 27 January at 18:00 UTC (Supplementary Figure S4a). Notably for the CSE of January 2016, the period when the JS was strengthening coincided with a steep consistent drop of 15 hPa in the AL MSLP, dipping from 1012 hPa on 20 January at 06:00 UTC to the month's lowest attained value of 997 hPa on 26 January at 00:00 UTC (Supplementary Figure S4b). The jet streak strengthened as it progressed eastward and extended longitudinally until it reached its maximum strength on 27 January 2016. This jet streak strengthening indicated a rapid progression of upper air cold winds from SH while the persistent pattern of AL intensification indicated convection in the atmospheric column underneath the jet streak.

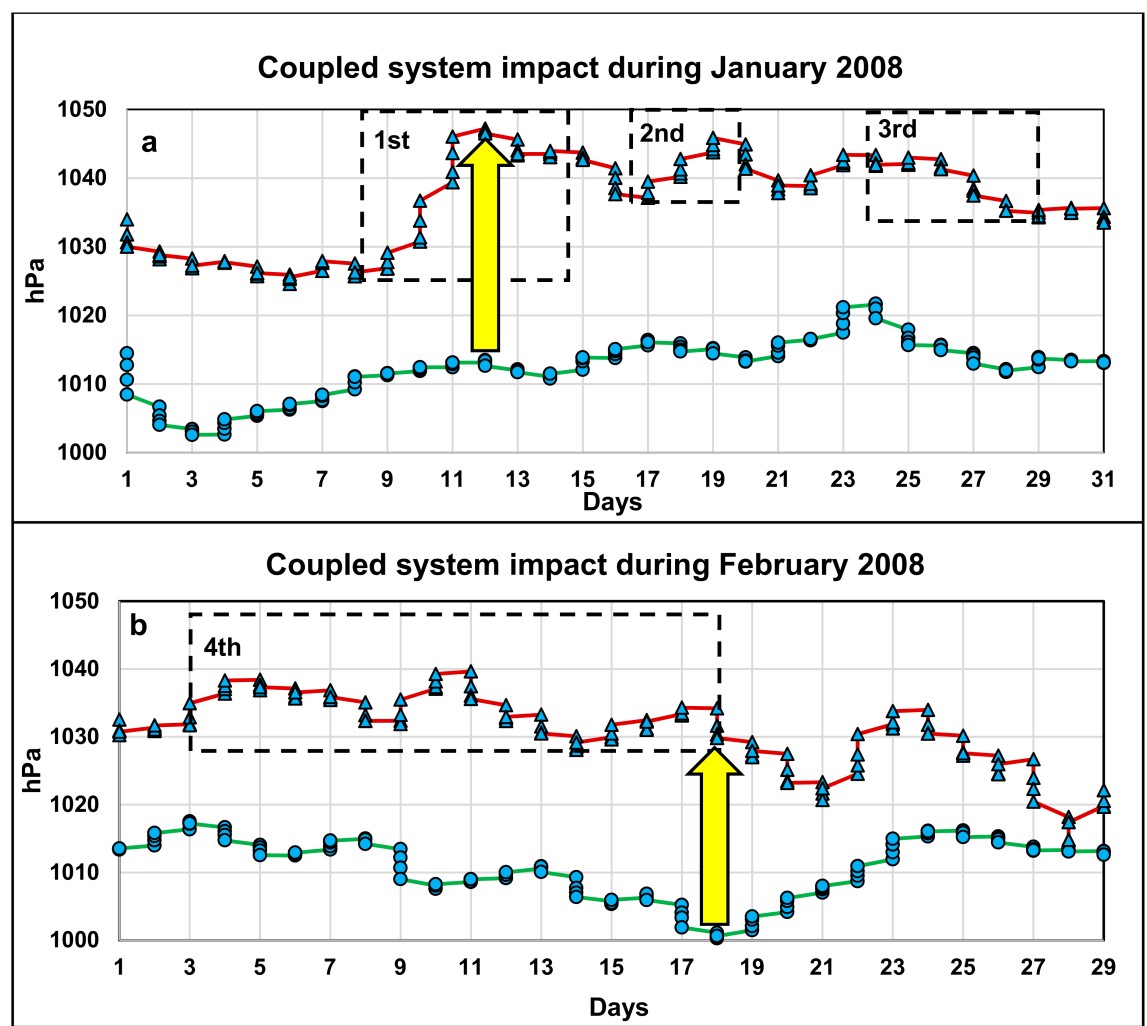

**Figure 5.** MSLP over the SH and AL domain for (**a**) January 2008 and (**b**) February 2008 computed at 00:00, 06:00, 12:00, and 18:00 UTC. Red lines in (**a**) and (**b**) indicate intraday MSLP over the SH and green lines in (**a**) and (**b**) indicate intraday MSLP over the AL. Yellow arrows in (**a**) and (**b**) indicate the maximum PD between SH and AL systems attained during the month. "1st", "2nd", "3rd" and "4th" in the figure represent the categorized CSE episodes. Data from the ECMWF, ERA-Interim.

For the CSE during January–February 2008, a very distinct pattern of zonal JS as an elongated continuous stretch of JS extending from 70 °E to 180 °E is observed (Figure 6a). This pattern originated on 26 January 2008 at 00:00 UTC and continued nearly for 20 days remaining until 15 February at 12:00 UTC (Figure 6b), largely overlapping with the HKO recorded cold spell duration of 24 January to 16 February 2008. This JS later started to split from 15 February 2008 at 18:00 UTC and further weakened as it moved eastward.

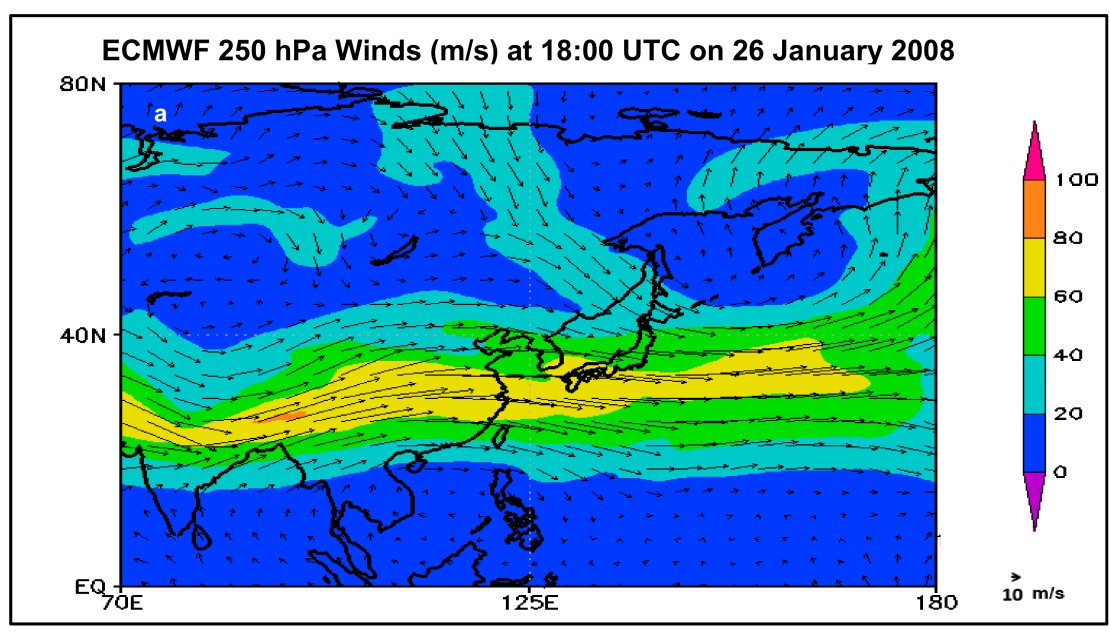

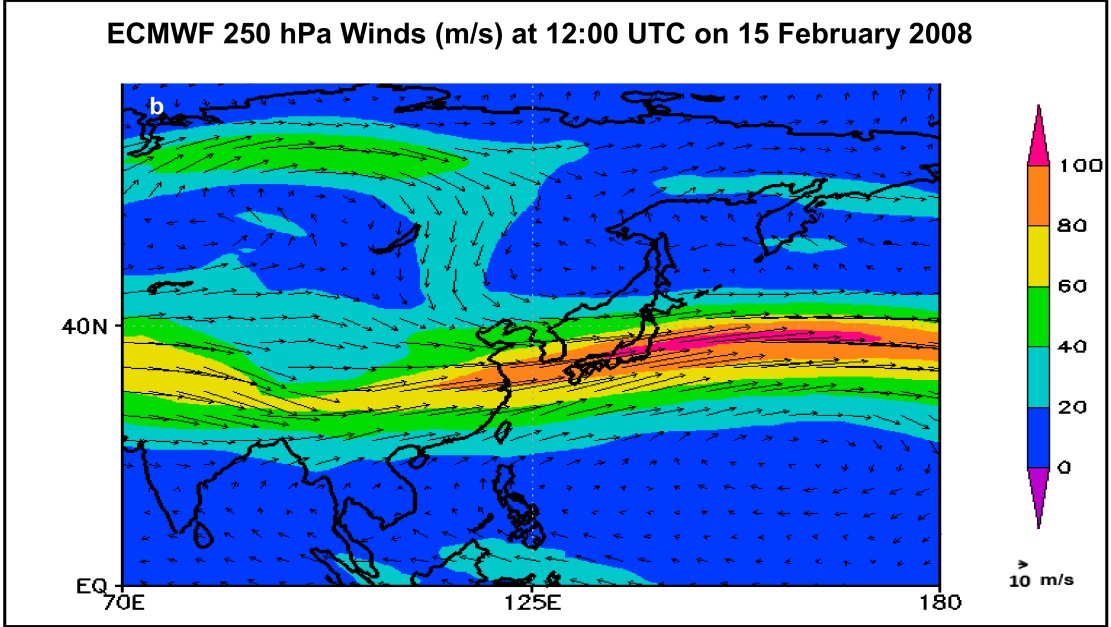

**Figure 6.** JS over East Asia towards Northern Pacific at 250 hPa (**a**) at 18:00 UTC 26 January 2008 and (**b**) at 12:00 UTC 15 February 2008. Data from the ECMWF, ERA-Interim.

## 3. A New Coupling Framework

To summarize, we investigated the SH–AL intensity, large-scale mid-latitude PD and structure of JS that links CSE to the large-scale circulation patterns in the atmosphere. To support the development of appropriate diagnostics for detecting onset of unusual CSEs, a generalized description of the key meteorological mechanisms that cause CSE in East Asia is shown schematically in Figure 7.

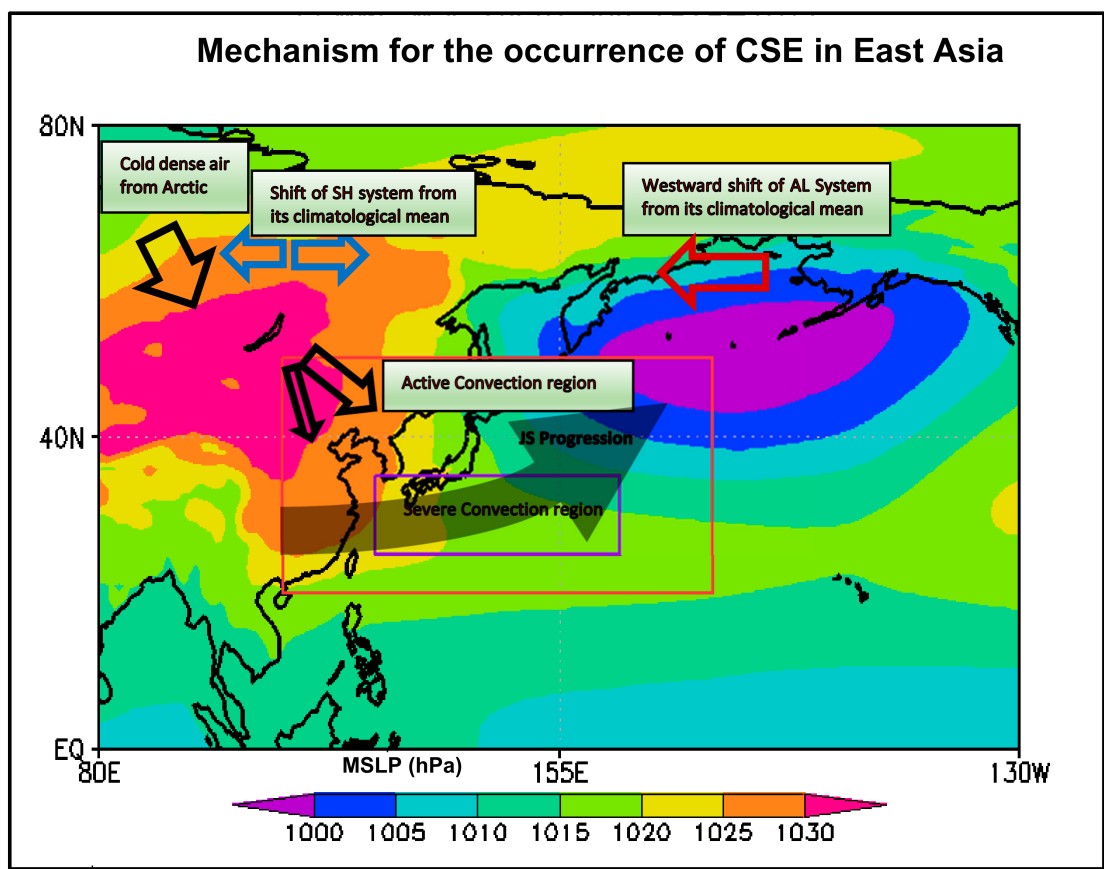

**Figure 7.** The progression of continental polar air masses (indicated by a black thick arrow) from Arctic towards Siberia as evidenced through strengthening of the MSLP. The cold air masses as it reaches Mongolia bifurcates into two major tracks (indicated by two black arrows). The progression of the Jet Stream (JS) is indicated by a thick grey-shaded arrow within the red box. The red box in the figure indicates the region of active convection during December, January, and February while the violet box indicates the region of strong convection during January. Red arrow indicates the shifting of AL system westward from its climatological averaged center. Blue arrows indicate the shifting of SH system eastward/westward from its climatological averaged center.

The intensification of PD drives copious amounts of cold dry air from Arctic Circle towards East Asia and South China, reaching far into the SCS. The strongest PD over period of 38 years (1979–2016) is observed for January 2016 (i.e., during the CSE of January 2016). The upper tropospheric flow is characterized by the strengthening of the jet streak within the JS. The eastward progression of the strengthened jet streak leads to intense divergence of air masses at upper air column. It further leads to intense short-lived (24–48 hour) low pressure systems (LPSs) forming frequently between the coast of Japan and the western Pacific (indicated as active convection within the red box for December, January, and February and as strong convection within the violet box for January). The annual mean vertical velocity within the active convection region (indicated in Figure 7) between 1000 hPa and 250 hPa peaks at 900 hPa to a maximum in January (0.023 Pa/s) followed by December (0.022 Pa/s), and February (0.020 Pa/s) (Supplementary Figure S5) while the vertical cross section at Lat 40 °N within the active convection region shows the continuous ascending and descending winds from the surface to 250 hPa for December, January, and February (Supplementary Figure S6). This further shows that convection within the region is typically more intense near the coast of Japan with the ascending winds reaching an altitude of 250 hPa at 0.1 Pa/s and descending winds reaching the surface at (−) 0.2 Pa/s during December, January, and February.

The annual zonal wind fields at 250 hPa during the same period shows a varying wind fields from 10 m/s to 50 m/s over the convective region between 40 °N and 50 °N due to passage of the Jet Stream (Supplementary Figure S7 for wind fields at 250 hPa). For example, the presence of a jet streak at 250 hPa (between 70 and 90 m/s) led to the formation of frequent LPS during January 2016 as readily observable for 11 days, specifically, 1, 5, 7, 11, 12, 13, 15, 19, 20, 25, and 31 January (see Supplementary Figure S8 for typical MSLP plots). During December 2009, the LPS is observed over 17 days (2–31 December; see Supplementary Figure S9 for typical MSLP plots), while for Januart–February 2008 it is observed over 12 days (6 January to 13 February; see Supplementary Figure S10 for typical MSLP plots). The formation of these LPS as determined by the strength and pattern of jet streak leads to the bifurcation of CSE tracks over Mongolia i.e., moving towards western pacific or towards SCS. Associated with the jet streak, we found a continuous long JS stretching all the way from southwest China to northern Pacific, indicating a large upper air temperature difference. This consistent pattern associated with JS remained stagnant for days, leading to colder than average weather over south central China as evidenced in the January–February 2008 CSE. Furthermore, the weather situation during this period drastically worsened with the longitudinal shift of the AL system westward from its climatological mean. This westward shift allowed higher dominance of AL system over the surface winds originating from the SH system resulting in winds streaming into China from the west rather than the usual northeasterly route. These together lead to the unprecedented CSE reported over January–February 2008.

This new coupling framework on the synoptic drivers of CSEs enables a more fundamental understanding of CSE onset based on the linkages beyond the current numerous region-specific criteria adopted. We further propose a new meteorological CSE scale for identifying the intensity and direction of CSE in East Asia shown in Table 1 based on the monthly averaged PD intensity and jet streak strength with respect to their climatological mean. To be classified as strong CSE, a CSE must have a SH strength of at least 1030 hPa and PD intensity of 19 hPa. The higher classification of extreme CSE has SH strength exceeding at least 1036 hPa and PD exceeding 29 hPa. Furthermore, the scale also provides a guidance on the progression of CSE. A jet streak value ≤67 m/s leads to CSE progressing towards SCS while a higher value between 68 and 77 m/s leads to the bifurcation of CSE track towards SCS and Japan. The highest value of ≥78 m/s majorly drives the CSE track towards Japan. Validation of the scale is performed using reported 11 unusual CSEs that resulted in severe socio-economic disasters as reported by EM-DAT and augmented with information from the Japan Meteorological Agency, the Korean Meteorological Administration, and the HKO (Table 2). For all these reported CSEs, the SH is found to be higher than its climatological mean for all years except 2009. Similarly, the AL is found to be lower than its climatological mean except January 2008. The higher jet streak values indicate intense upper level divergence leading to atmospheric convection at the surface, triggering formation of the LPS. More importantly, the PD being higher than its climatological mean indicates a strong and persistent pressure systems leading to CSEs. Notably the January–February 2008 CSE, in addition to having higher SH and AL strengths and PD intensity value, also had the highest westward longitudinal shift of $22.5^0$ of the AL from its climatological mean, allowing abundance of cold air to stream into mainland China, leading to the CSE to be unprecedented in duration and spatial extent.

**Table 1.** The CSE-PD scale criteria for classifying cold surge events (CSEs) and their progression based on monthly averaged PD (hPa) and jet streak (m/s) from 1979 to 2016. SCS: South China Sea.

| Category | SH Strength (hPa) | PD Intensity (hPa) | Direction of CSE Jet streak (m/s) | |
|---|---|---|---|---|
| Strong CSE | 1030–1035 | 19–28 | ≤67 | towards SCS |
| | | | 68–77 | towards SCS, Korea, Japan |
| | | | ≥78 | dominance towards eastern China, Korea, Japan |
| Extreme CSE | 1036–1041 | 29–38 | ≤67 | towards SCS |
| | | | 68–77 | towards SCS, Korea, Japan |
| | | | ≥78 | dominance towards eastern China, Korea, Japan |

**Table 2.** Reported CSEs available from the Emergency Events Data Base (EM-DAT), categorized based on the CSE Scale of Table 1. High and low values in parentheses indicate value above and below the climatological mean. In the table 'NA' refers to data not available.

| Reported CSE | Category of CSE based on CSE Scale | Countries affected | Total Loss (USD, '000) | Strength SH (hPa) | Intensity PD (hPa) | Jet streak wind speed (m/s) | Westward Shifting of AL System (degrees) |
|---|---|---|---|---|---|---|---|
| January 2016 | Extreme CSE | China, Japan, Korea, Taiwan, Mongolia, Hong Kong | NA | 1037 (High) | 33.22 (High) | 69.66 (low) | No |
| February 2012 | Strong CSE | China, Japan | 20,200 | 1031 (High) | 21.9 (High) | 74.62 (High) | No |
| January 2011 | Extreme CSE | China, Korea, Japan | 281,000 | 1040 (High) | 30.67 (High) | 82.00 (High) | Yes, by 10 deg. |
| December 2009 | Strong CSE | Mongolia, Hong Kong | 62,000 | 1029 (Low) | 19.39 (High) | 60.16 (Low) | No |
| January 2008 | Strong CSE | China, Hong Kong | 21,100,000 | 1036 (High) | 24.52 (High) | 68.75 (Low) | Yes, by 22.5 deg. |
| February 2008 | Strong CSE | | | 1031 (High) | 20.55 (High) | 62.52 (Low) | No |
| December 2005 | Extreme CSE | Japan | NA | 1037 (High) | 31.38 (High) | 78.01 (High) | No |
| December 2001 | Strong CSE | Taiwan | NA | 1036 (High) | 26.57 (High) | 65.37 (High) | No |
| January 1981 | Extreme CSE | China, Korea, Japan, | NA | 1033 (High) | 32.45 (High) | 81.28 (High) | No |
| February 1968 | Strong CSE | China, Hong Kong | NA | 1036 (High) | 28.52 (High) | 65.65 (Low) | No |
| January 1984 | Strong CSE | Japan | NA | 1033 (High) | 21.76 (High) | 77.78 (High) | No |

## 4. Summary

The current use of different location-specific criteria for interpreting CSE onset does not facilitate understanding of the synoptic relationships between atmospheric circulation and surface conditions for CSE onset. Furthermore, unusual CSEs affecting East Asia encompass a wide spatial extent, making such location-specific metrics difficult to interpret. A way forward is to investigate and characterize

past unusual CSEs with respect to the underlying synoptic processes for improving our understanding for CSE onset. A framework of likely pathways is developed here (summarized in Figure 7) for linkages over monthly timescales between the SH and AL leading to large-scale mid-latitude PD intensity and along with the structure of the JS in characterizing the processes causing unusual CSEs in East Asia. Table 1 further represents a meteorological scale for identifying the strength and propagation direction of such CSEs. Under the proposed framework, we conclude that only a prolonged and enhanced PD as developed due to the strengthening of SH system, intensification of the coupled AL system, and under continued influence from JS, results in unusual CSEs in East Asia. Our conclusion does not exclude the possibility of further contributions from additional mid-latitude or high-latitude processes in improving the framework for CSE onset. Such an assessment would be beneficial towards weather predictions supporting impact and risk-based forecasts, given the potential negative impacts of such unusual CSEs.

**Supplementary Materials:** The following are available online at http://www.mdpi.com/2225-1154/7/2/30/s1, Figure S1: Differences in SH intensification for (a) December; (b) January; and (c) February during the period 1979-2016. Data from the ECMWF ERA Interim, Figure S2: ECMWF, ERA Interim plots for MSLP at 00 UTC on 22nd Jan 2016 showing formation of intense pressure gradient between SH and AL systems, Figure S3: Differences in JS intensity (m/s) at 250 hPa for (a) December; (b) January; and (c) February during the period 1979–2016, Figure S4: (a) Jet Streak intensity at 250 hPa (m/s) and (b) AL intensity (hPa) for Jan 2016 at 00, 06, 12, 18 UTC, Figure S5: Vertical velocity (Pa/s) over the active convection region for Dec, Jan and Feb for the period 1979-2016 from 1000 hPa to 250 hPa computed from ECMWF, ERA Interim, Figure S6: Vertical winds (Pa/s) at from 1000 hPa to 250 hPa along the cross section at 40 0N over active convective region for (a) Dec; (b) Jan; and (c) Feb for the period 1979-2016 computed from ECMWF, ERA Interim, Figure S7: Climatology of winds (m/s) at 250 hPa over active convective region for (a) Dec; (b) Jan; and (c) Feb for the period 1979–2016 computed from ECMWF, ERA Interim, Figure S8: ECMWF, ERA Interim plots for MSLP at 00 UTC for (a) 11th Jan 2016; (b) 15th Jan 2016; (c) 19th Jan 2016; and (d) 25th Jan 2016 showing formation of frequent LPS during Jan 2016 between 140 °E–180 °E and 30 °N–55 °N, Figure S9. ECMWF, ERA Interim plot for MSLP at 00 UTC for (a) 7th Dec 2009; (b) 14th Dec 2009; (c) 18th Dec 2009; and (d) 22nd Dec 2009 showing formation of frequent LPS during Dec 2009 between 140 °E–180 °E and 30 °N–55 °N, Figure S10: ECMWF, ERA Interim plot for MSLP at 00 UTC for (a) 10th Jan 2008; (b) 3rd Feb 2008; (c) 8th Feb 2008; and (d) 13th Feb 2008 showing formation of frequent LPS during Jan–Feb 2008 between 140 °E–180 °E and 30 °N–55 °N, Table S1: Surface temperature anomalies over 2001–2010 with respect to 1961–1990 globally, and in the northern hemisphere and southern hemispheres along with warmest/least warm year during 2001–2010, and warmest/coldest decade over 1881–2010 (adopted from WMO, 2013), Table S2: The World's warmest year on record (adopted from WMO, 2017).

**Author Contributions:** A.K., E.Y.M.L., and A.D.S. designed the problem and solution approach. All authors contributed to the writing of the manuscript.

**Funding:** This research received no external funding.

**Acknowledgments:** The study was supported by the Interdisciplinary Graduate School and the Institute of Catastrophe Risk Management, both at the Nanyang Technological University, Singapore.

**Conflicts of Interest:** The authors declare no conflict of interests.

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
