# Peer review of "Relationship between East Asian Cold Surges and Synoptic Patterns: A New Coupling Framework"

_climate, doi:10.3390/cli7020030_

Round 1

Reviewer 1 Report

The revised manuscript “Relationship between East Asian cold surges and synoptic patterns: A new coupling framework” by Anupam Kumar, Edmond Y.M.Lo and Adam D.Switzer addressed most of my concerns of the manuscript in the last version. Figure 2 was improved. The authors give a further discussion on the progression of the intense cold air masses from higher towards lower latitudes resulting in an enhanced pressure difference (PD) and driving the winds from the Arctic Circle through Eastern Russia and towards the Aleutian Low (AL) system. And the mechanism driving the East Asian cold surge Events (CSEs) was further discussed. But I still have the following questions that expected to be answered before the manuscript can be accepted for publication in Climate.

Specific points

1. From Supplementary Figure 5, it shows the descending motion (max at 0.023 Pa/s in Jan; 0.022 Pa/s in Dec; 0.020 Pa/s in Feb, respectively) in the region of red box, which was defined as the “active convection” field (in Figure 7) from 1000 hPa to 250 hPa during boreal winter season. It seems that there is only descending motion and no ascending motion in the region of red box. Please give more explanations on the “active convection” over the region of red box.

Author Response

The revised manuscript “Relationship between East Asian cold surges and synoptic patterns: A new coupling framework” by Anupam Kumar, Edmond Y.M.Lo and Adam D.Switzer addressed most of my concerns of the manuscript in the last version. Figure 2 was improved. The authors give a further discussion on the progression of the intense cold air masses from higher towards lower latitudes resulting in an enhanced pressure difference (PD) and driving the winds from the Arctic Circle through Eastern Russia and towards the Aleutian Low (AL) system. And the mechanism driving the East Asian cold surge Events (CSEs) was further discussed. But I still have the following questions that expected to be answered before the manuscript can be accepted for publication in Climate.

Specific points

1. From Supplementary Figure 5, it shows the descending motion (max at 0.023 Pa/s in Jan; 0.022 Pa/s in Dec; 0.020 Pa/s in Feb, respectively) in the region of red box, which was defined as the “active convection” field (in Figure 7) from 1000 hPa to 250 hPa during boreal winter season. It seems that there is only descending motion and no ascending motion in the region of red box. Please give more explanations on the “active convection” over the region of red box.

[Response:] Yes, ascending and descending motions are not captured in Supplementary Figure 5 as it shows the climatological vertical velocity at various pressure levels during Dec, Jan and Feb as spatially averaged over the active convection region (red box). For showing both ascending and descending motions a new figure (Supplementary Figure 6) showing cross sections within the convective region at latitude 400 N is added. Ascending and descending motions are clearly seen. Elaboration on the resulting surface convergence and divergence is provided in line L318-322 in the revised manuscript.

Reviewer 2 Report

“Jet Streak” should precede “wind speed” in Table 2.

Author Response

Jet Streak” should precede “wind speed” in Table 2.

[Response:] The relevant column heading in Table 2 is ‘Jet streak wind speed (m/s)’ with ‘Jet streak’ preceding ‘wind speed’.

Reviewer 3 Report

Minor Comments: 

L13:  keep distinct or unusual, but not both.

L14: strike affected

L23-25: Sentence unclear, ‘over a month synoptic patterns’?  I would state the number of cases first. E.g. “Based on three unusual and eight strong-extreme CSEs, …”

L31: Since you are primarily discussing the impacts of specific events, and not climate change in this paragraph, the paper doesn’t directly deal with climate change, I would strike this sentence and start with the 2nd sentence. The original sentence could potentially be incorporated into the 2nd paragraph which does discuss climate change.

L54:  comma after SH

L72: Single should not be capitalized.  ‘a single’?

L79: ‘The combined effect…’  I don’t understand the purpose of this sentence.  What are more severe local weather situations? (this phrase is also used on L199)

L84: What does unusual mean in this context? Do you mean strong?

L110: I still don’t think this is a ‘substantial trend’, at least as presented. Only one month has a p value of 0.06 which means none of the months fall within the standard of significance (p<=0.05). My personal opinion is the tone of the language should be brought down some. That’s not to say I don’t think there is an intensification here—just that it’s important to express it correctly.

Figure 2: the panel letters are still hard to read. I would make these larger and in black.

L162 (and other places): Is Coupled System a formal phrase? Note that only one word is capitalized here, but both are in other places.

L196: Strike ‘Please see’

Figure 3 caption: How about:  ‘ERA Interim PD (hPa) for Jan 2016 between the SH and AL domains computed in 6-hourly intervals.’  Strike dashed line (not discussed)

Figure 4: Note that red/green arrows are indistinguishable to color blind individuals. One color should be modified.

L211: that features a series of surges?

L212: reword sentence-  ‘ A recent study suggested this CSE was tied to… ‘?

Figure 7: Given that this is the key figure in the paper, I’d still encourage the consideration of a graphic artist to create a nice looking schematic overlaid on top of the MSLP pattern. For now, I maintain that this figure looks more like a powerpoint slide. Perhaps this is something the journal can assist with.

L309: severe active convection is an awkward phrase.  What is the definition of severe?

L315: Mention what level the jet streak is at (I’m also unsure why this is capitalized—this follows the thread of inconsistent/strange capitalization throughout the paper).

Other comments:

Sorry to hear that a number of these issues were due to the reformatting of the manuscript. Regarding GrADS, there isn’t anything inherently wrong with this—my overall opinion is the figures have a dated look compared to what might be possible with other plotting packages. As long as the editors are fine with the quality, it’s OK by me. Finally, I tried to find the minor grammatical issues as listed above. I consider myself pretty poor at this process, however, so I would encourage the journal to have a English copy editor go through and assist with cleaning up things I may not have caught.Parts of the text are still speculative based off the sample of cases made. This comment was not addressed in the responses.

Author Response

Minor Comments: 

L13:  keep distinct or unusual, but not both.

[Response:] I have kept “distinct” and removed “unusual”.

L14: strike affected

[Response:] I have removed the word “affected”

L23-25: Sentence unclear, ‘over a month synoptic patterns’?  I would state the number of cases first. E.g. “Based on three unusual and eight strong-extreme CSEs, …”

[Response:] I have revised the sentence as suggested (line 21) but further improved lines 21-25 for clarity.

L31: Since you are primarily discussing the impacts of specific events, and not climate change in this paragraph, the paper doesn’t directly deal with climate change, I would strike this sentence and start with the 2nd sentence. The original sentence could potentially be incorporated into the 2nd paragraph which does discuss climate change.

[Response:] I have removed the sentence “It is reported that climate change is “perhaps the single greatest challenge confronting the Asia-Pacific region, and its more than 4 billion people1”. The sentence is moved to new line 48-49.

L54:  comma after SH

[Response:] I have added comma after SH in new line L56.

L72: Single should not be capitalized.  ‘a single’?

[Response:] Single replaced by ‘a single’ in new line L74.

L79: ‘The combined effect…’  I don’t understand the purpose of this sentence.  What are more severe local weather situations? (this phrase is also used on L199)

[Response:] Upon re-examination, we have deleted this sentence since it does not add to the context of the coupling framework being discussed.   

L84: What does unusual mean in this context? Do you mean strong?

[Response:] Yes it means strong CSEs, a subset of which is further classified as extreme based on the proposed meteorological scale. The word ‘unusual’ is replaced by ‘strong’ in this sentence.

L110: I still don’t think this is a ‘substantial trend’, at least as presented. Only one month has a p value of 0.06 which means none of the months fall within the standard of significance (p<=0.05). My personal opinion is the tone of the language should be brought down some. That’s not to say I don’t think there is an intensification here—just that it’s important to express it correctly.

[Response:] We have changed the word ‘substantial’ to ‘noticeable’ in line L114 and the word ‘significant’ to ‘noticeable’ in L127.

Figure 2: the panel letters are still hard to read. I would make these larger and in black.

[Response:] The font size of the panel letters (a, b, c, and d) in Fig. 2 is increased from 10 to 12 for better readability. Changing to black font actually deceases the readability and as such we are not done so.

L162 (and other places): Is Coupled System a formal phrase? Note that only one word is capitalized here, but both are in other places.

[Response:] The notation has been changed consistently to ‘Coupled system’ in heading of Section 2.2 (Line 166) and in the legends of Figure 5.

L196: Strike ‘Please see’                                                    

[Response:] We have removed the words ‘Please See’, in new line L200.

Figure 3 caption: How about: ‘ERA Interim PD (hPa) for Jan 2016 between the SH and AL domains computed in 6-hourly intervals.’  Strike dashed line (not discussed)

[Response:] As suggested, we have changed the caption ( L208-209). The dashed line in Figure 3 is deleted.

Figure 4: Note that red/green arrows are indistinguishable to color blind individuals. One color should be modified.

[Response:] The size of the arrows are increased and green is replaced by orange.

L211: that features a series of surges?

[Response:] This is modified as suggested ( L216).

L212: reword sentence-  ‘ A recent study suggested this CSE was tied to… ‘?

[Response:] As per the suggestions, we have corrected this in L217.

Figure 7: Given that this is the key figure in the paper, I’d still encourage the consideration of a graphic artist to create a nice looking schematic overlaid on top of the MSLP pattern. For now, I maintain that this figure looks more like a powerpoint slide. Perhaps this is something the journal can assist with.

[Response:] Thank you for your suggestions. Given the time allowed for revisions, we will defer to the Journal office. However, we believe that the figure should suffice in presenting the key concepts in the proposed framework.

L309: severe active convection is an awkward phrase.  What is the definition of severe?

[Response:] To differentiate from ‘active convection’ in L314 we have now used the term ‘strong convection’ (L315 and L301-302) instead of previously used ‘severe active convection’.

L315: Mention what level the jet streak is at (I’m also unsure why this is capitalized—this follows the thread of inconsistent/strange capitalization throughout the paper).

[Response:] The level of jet streak at 250 hPa is now mentioned in new line L324. The word “Jet Streak” is now entirely replaced by the word “jet streak” throughout the manuscript.

Specific points:

1. Figure 2 was not clearly noted. Which one is Figure 2a? And which is Figure 2c? The figure in the last panel is not clear.

[Response:] We have increased the font size of the legends (a, b, c, d) from 10 to 12 in the figure to increase the readability. Figure 2a is in the top left panel and while figure 2c is the bottom left panel.

Thank you for your time and consideration.

This manuscript is a resubmission of an earlier submission. The following is a list of the peer review reports and author responses from that submission.

Round 1

Reviewer 1 Report

See attachment for comments.

Reviewer 2 Report

This is a very interesting research about the relationship among the Siberian High, the Aleutian Low, the Jet Stream, and the Cold Surge Events affecting East Asia. Although three specific cases are presented, one figure explaining the mechanism for the occurrence of these events is introduced, and one classification is proposed and applied to eleven events in around forty years. Consequently, the paper merits to be published after the introduction of some changes. The main drawback is that the reason of choosing the three events selected, L 81, should be introduced.

Minor remarks.

L. 15. References in the abstract should be avoided.

L. 33. Acronyms appearing only twice, such as NEM, should not be suggested to simplify the text.

L. 91. Revise the sentence. “Many previous analyses for CSE have been “

L. 109. Revise pressure, since the figure indicates 1035 hPa, and year, 1988, instead of 1998.

L. 118. Revise the sentence “…the presence of heat trapping clouds…”.

Figure 2 should be revised since some labels are missing. Certain texts are written with varied fonts and incomplete, and one figure is unclear. Moreover, the caption refers to white arrows. However, there is only one arrow in the figure.

L. 186. Edit sentence. “Here it seen”may be replaced by “Figure 3b shows that progression…”

Figure 5 should be improved. Introduce b in the second figure, which is shifted, and remove the rest of the text unreadable.

Table 2. Does ‘000 mean thousands of dollars? Moreover, replace “Direction Jet Streak” by “Jet Streak wind speed”.

L. 399. Replace “asso-ciated” by “associated”

Reviewer 3 Report

The manuscript “Relationship between East Asian cold surges and synoptic patterns: A new coupling framework” by Anupam Kumar, Edmond Y.M.Lo and Adam D.Switzer investigates the strong Cold Surge Events (CSEs) occurrences in East Asia by analyzing the relationship between large-scale synoptic patterns and the potentially influence by changes in the Arctic. The results show that the coupling of Siberian High (SH) and Aleutian Low (AL) can make the pressure difference (PD) between East Asian continent and north Pacific Ocean increase and then results in the outbreak of the unusual CSE. The result is interesting and meaningful to understand the CSE phenomenon. It is suggested to be accepted for publication in Climate with major revision. Specific points: 1. Figure 2 was not clearly noted. Which one is Figure 2a? And which is Figure 2c? The figure in the last panel is not clear. 2. In Figure 6, the redline rectangle region shows active convection region, could the authors calculate the vertical velocity in this region? And show the divergence field over 250 hPa ? 3. The abstract of the manuscript does not give the main results derived, please rewrite it. 4. Line 186 “…… progression of the intense cold air masses from higher towards lower latitudes is due to the intensification of AL, resulting in an enhanced PD and driving the winds from the Arctic Circle through Eastern Russia and towards the AL system.” Could the authors explain why the progression of the cold air masses from higher latitudes is “due to the intensification of AL”? What the mechanism is that?